# [Re] Generative causal explanations of black-box classifiers

## Reproducibility Summary

Explainability of black-box classifiers is an important aspect of neural models that often is non-existent. Classifiers made for tasks such as object recognition and decision making often lack transparency which causes vulnerability being overlooked [1]. Without insight into the reasons behind a decision made by a neural model, potential security risks or classification mistakes can be missed [1]. Multiple solutions have been posed to solve this problem. An example is the method designed by O'Shaugnessy et al. [2]. The authors design a learning framework that leverages a generative model and information-theoretic measures of causal influence. The objective function encourages both the generative model to faithfully represent the data distribution and the latent factors to have a large causal influence on the classifier output. In this study, the reproducibility of the method developed by O'Shaugnessy et al. is tested. Several claims are challenged to ensure the validity of the method. Furthermore, the method is extended to test generalizability. It was found that the claims are not as strong as the authors suggested and the method is not as easily generalizable as expected. However, for the task described in the original study, the method is completely reproducible, and thus a valid contribution to machine learning innovation.

# 1    Scope of reproducibility

In "Generative causal explanations of black-box classifiers" the authors propose an explanatory model that can explain any black-box classifier post-hoc based on a learned low-dimensional representation of the data [2]. They call this explanatory model a generative causal explainer (GCE). The authors have designed a framework that leverages a generative model combined with an information-theoretic measure of causal influence. They use these by creating an objective function that encourages latent factors in the generative model to represent the data distribution and to have a large causal influence on the classifier output. The main claim of the paper is that their method can generate explanations for any classifier that admits class probabilities and a gradient.

To review this paper and its central claim we will attempt to reproduce parts of the paper, and go beyond the original results by carrying out extra experiments. Reproducing the figures presented in the original paper was quite easily done using the authors provided code. Because of this the main focus of our research will be based on extra experiments where we more thoroughly investigated the method that is proposed in the paper. The reproduction of the figures can be found in Appendix D.2.

The extra experiments are grouped in three main parts. (1) The central claim will be tested by assessing the method with different classifiers with architectures based on competitors of the ImageNet challenge [3]. (2) We investigate more thoroughly the parameter selection of the paper. In the paper the authors propose 3 parameters that dictate the number of causal and non-causal variables, and their influence on the objective function. These parameters are denoted by K, L and $\lambda$. In the original method the authors provide and use a heuristic algorithm to determine the optimal configuration of these model parameters. Although this heuristic algorithm is described, the authors do not state these results, and therefore do not show the sensitivity of the model to parameter choice. Therefore, varying the parameters will shed light on this sensitivity. (3) The generalizability of the model will be investigated by testing the method with more complex datasets than the one used in the original paper.

In summary, the following will be tested.

- The claim that the method works with any gradient-based black-box classifier.

- The influence of the parameters K, L and $\lambda$ on the explanation performance.

- The generalizability of the method to a more complex dataset.

# 2    Methodology

## 2.1    Causal Model Description

In the paper the authors attempt to causally explain black-box classifiers. Explanations, in this case, take the form of a low-dimensional and independent set of "causal factors" $\alpha \in \mathbb{R}^K$. Therefore, manually changing these parameter values should produce a corresponding change in the classifier output statistics. The method also allows for additional independent non-causal factors $\beta \in \mathbb{R}^L$. These factors do not change the corresponding output of the model. Together, these factors $(\alpha, \beta)$ form a low-dimensional representation of the real data distribution of a given dataset $X$. In the paper it is described that a key feature of these latent factors $(\alpha, \beta)$ is their independence. This independence allowed for their chosen metric for causal influence to simplify to the mutual information metric.

The generative model that is used to form the low-dimensional explanatory factors is mainly a variational auto-encoder (VAE). But the authors also propose analysis using a linear-Gaussian generative map, this map is mainly used to showcase geometric intuition that illuminates the function of their proposed training objective (described in Section 2.1.2).

### 2.1.1    Causal Influence Metric

The authors decide on a causal influence metric called information flow. Information flow quantifies the causal influence of observational distributions in the standard definition of conditional mutual information with interventional distributions. Due to the independence of $\alpha$ and $\beta$ the information flow from $\alpha$ to $Y$ coincides with the mutual information between $\alpha$ and $Y$.

$$I(\alpha; Y) = \mathbb{E}_{\alpha, Y} \left[ \log \frac{p(\alpha, Y)}{p(\alpha), p(Y)} \right] \tag{1}$$

### 2.1.2 Optimization Objective

To learn the generative mapping that explains a given black-box classifier the authors construct an objective function that enables the mapping to reconstruct the original data distribution, ensures independence between $\alpha$ and $\beta$, and dictates a large causal influence of $\alpha$ on $Y$. The objective function is defined as:

$$\arg\max_{g \in G} C(\alpha, Y) + \lambda \cdot \mathcal{D}(p(g(\alpha, \beta)), p(X)) \tag{2}$$

where $g$ is a generative function (in our case a VAE), $C(\alpha, Y)$ is the metric for causal influence from Eq. 1, and $\mathcal{D}$ is a measure of similarity between the generative function and the actual dataset. The $\lambda$ parameter controls how strongly the model should represent the actual underlying dataset. Careful selection of this parameter is required to ensure that the distribution $p(X|\alpha, \beta)$ lies in the data distribution $p(X)$, but this similarity term $(\lambda \cdot \mathcal{D}(p(g(\alpha, \beta))))$ should not overwhelm the causal influence term.

### 2.1.3 Training Procedure

The objective described in Eq. 2 is maximized using Adam. The causal influence term is computed using a sample-based estimate. The generative model uses a latent vector of length $K + L$ to generate new images that lie in the original data distribution. To estimate the causal influence term the first $K$ terms are sampled $N_\alpha$ times and the last $L$ terms are sampled $N_\beta$ times. An intuition behind this sample-based approach can be found in the original paper in Appendix D.

To train the causal explanatory model the parameters $K, L$, and $\lambda$ must be selected. Respectively they denote the number of causal terms, the number of noncausal terms, and the trade-off between causal influence and data fidelity in our objective. In the paper these parameters are tuned using a heuristic method shown in Figure 1. The authors do not expand on this selection method, nor do they investigate the effect of different selections of these parameters.

---

**Algorithm 1** Principled procedure for selecting $(K, L, \lambda)$.

1: Initialize $K, L, \lambda = 0$. Optimizing only $\mathcal{D}$, increase $L$ until objective plateaus.
2: **repeat** increment $K$ and decrement $L$. Increase $\lambda$ until $\mathcal{D}$ approaches value from Step 1.
3: **until** $\mathcal{C}$ reaches plateau. Use $(K, L, \lambda)$ from immediately before plateau was reached.

---

Figure 1: Procedure for selection K, L and $\lambda$ [2].

## 3 Implementation

All implementation was done in Python 3.8 using Pytorch 1.7.1, and the models were trained on local machines [4]. The original code provided by the authors was written in pytorch [1]. All our code is avaliable from GitHub [2]. Further instructions on how to run our provided code can be found on the repository.

### 3.1 Datasets

The original datasets used were the well known MNIST and fashion MNIST (fMNIST) datasets [5, 6]. These datasets were supplied alongside the rest of the code provided by the authors. MNIST is a collection of written digits and fMNIST is a collection of clothing articles traditionally used for benchmarking machine learning algorithms. Both MNIST and fMNIST consist of 70.000 grayscale images with a resolution of 28 x 28 pixels. The paper did not apply any transforms on the dataset. For part of our experiments we used the CIFAR-10 dataset. CIFAR-10 consists of 60.000 labeled images that are subdivided into 10 classes, much like the MNIST dataset. The images have a resolution of 32 x 32 pixels, which is slightly larger than the images in the MNIST dataset. However, considering the complexity of the objects in the CIFAR-10 images, the resolution is relatively low. A resolution of 28 x 28 is sufficient for clearly displaying a written number, but displaying a truck with a resolution of 32 x 32 means that quite some detail is lost. Furthermore, the images are coloured. Resulting in two more channels that convey extra information about the object. CIFAR-10 is used without applying any transforms, with 50,000 training images and 10,000 testing images. For

---

[1] https://github.com/siplab-gt/generative-causal-explanations
[2] https://github.com/DanielPerezJensen/FACT-anonymous

training the classifier, the training set is split up in 40,000 training images and 10,000 validation images, allowing us to closely monitor the model performance.

## 3.2 Models

To reproduce the figures as shown in the original paper we retrained all models with the provided code and recreated the figures using the provided scripts.

As described in Section 1 we wanted to test the method more extensively by testing the method with more classifiers than the one provided in the authors' code. Furthermore we also wanted to test how the K and L scale with the complexity of the dataset and evaluate how the method performs on relatively more complex datasets. To test these points we created more classifier models described below and we needed more complex generative models that are able to represent the underlying data distribution.

### 3.2.1 Classifiers

To test the generalisability of the method we tested the method using different classifiers. The authors used a relatively simple shallow network to work with. For our own experiments we created and tested three architectures based on ResNet, DenseNet and InceptionNet [7, 8, 9]. All these models were state-of-the-art when proposed. All implementation details about these classifiers can be found on our code repository and the models are described in Appendix A.1.

### 3.2.2 Generative Models

**CVAE**    The authors provided two architecures in their code handed in alongside their work. They defined a Convolutional Variational Autoencoder (CVAE), a VAE consists of an encoder and a decoder. The encoder maps input to a lower-dimensional latent space, the decoder then takes this mapping and reproduces the output. The latent space follows properties that allow us to generate new samples [10]. Both the encoder and the decoder consist of three convolutional layers followed by three ReLU activation layers in the encoder and two ReLU activations and a final sigmoid layer in the decoder. The encoder ends with two separate linear layers for the mean and log-variance, the parameters of the latent distribution.

Using CIFAR-10 increases dimensionality compared to MNIST/fMNIST. CIFAR-10 consists of RGB images with each color channel having a range of 0-255. While MNIST is grayscale and uses one binary channel. Using such a dense dataset suggests the need of models with higher complexity. Hence we introduce the following variant of the CVAE:

**CVAEImageNet**    The CVAEImageNet is almost entirely the same as the original CVAE described above. The CVAEImageNet architecture was also defined by the authors in their provided code. The only difference is that each convolutional layer is followed by a batch normalisation layer. A batch normalisation layer standardises the inputs to its consecutive layer, which stabilises the learning process along with reducing the amount of epochs needed to obtain convergence. Batch normalisation is an approach to eliminate the phenomenon called internal covariate shift. The internal covariate shift is the effect of the input distribution shifting while the input is fed through each layer which causes the algorithm to chase a moving target. This is a common problem for advanced deep neural networks.

## 3.3 Hyperparameter Selection

To reproduce the figures in the original paper we used the same hyperparameters as provided in the authors' code and paper. In general the authors used a batch size of 64 and a learning rate of 0.0005. All models were trained using the Adam optimizer.

Furthermore the optimal values of $K$, $L$ and $\lambda$ depend on what dataset the experiment was run. For the MNIST dataset the authors used values of respectively 1, 7 and 0.05. For the fMNIST dataset respectively 2, 4 and 0.05 were given as optimal hyperparameters. As described in Section 2.1.3, the authors used a sample-based method to estimate the causal influence metric. For all models trained to create the figures in the original report they used 25 samples for $\alpha$ and 100 samples for $\beta$ ($N_\alpha = 25, N_\beta = 100$). The $\alpha$ and $\beta$ vakyes were found by the authors using the heuristic method described in Figure 1. The $N_\alpha$ and $N_\beta$ values were not elaborated on in the paper.

However, the parameters used in this heuristic method and the results leading to the optimal parameter configuration are not discussed by the authors. Therefore, for reproduction purposes, we attempt to implement Algorithm 1 (as shown in Figure 1) in Python. First, step 1 optimises only the data similarity part, denoted by $\mathcal{D}$, of the objective

function (2), while increasing L on every iteration. Note that this term $\mathcal{D}$ measures the similarity between the learned data distribution and the real dataset. We use 3000 training steps to train the explainer in each iteration. The optimal L is found when $\mathcal{D}$ plateaus. Intuitively, this finds the total number of latent factors needed to adequately represent $p(X)$. We defined the plateau criteria as met when the relative improvement of $\mathcal{D}$ in successive runs is smaller than 1%. However, measuring this relative improvement turned out to be difficult because of high variance in the results of $\mathcal{D}$ (detailed results can be found in figure 14 in appendix B). To accommodate to this situation, we measure the relative improvement of the last 500 training steps. Second, step 2 optimises the causal influence term $\mathcal{C}$ of objective 2 while keeping $\mathcal{D}$ as optimal as possible. In each iteration L is decremented by 1 and K is incremented by 1, keeping the total number of latent factors equal. Per configuration of K and L, the optimal $\lambda$ term is determined, which functions as a trade-off term between causal influence and data fidelity. The optimal value is derived by step-wise incrementing $\lambda$ by 0.1 until $\mathcal{D}$ 'approaches' the optimal value from step 1, as described in figure 1. These two steps are repeated until $\mathcal{C}$ plateaus. In the implementation, the criteria for $\mathcal{C}$ plateauing is set to a relative improvement smaller than 1%. However, the criteria for $\mathcal{D}$ 'approaching' the optimal value from step 1 was more difficult as it has a large influence on the optimal $\lambda$ parameter will be chosen. Results of varying this criteria are discussed in the results section 4.2.

## 4 Results

### 4.1 Classifiers

The figures as they were originally presented in the paper can be found in Appendix 5.3. Furthermore we reproduced these figures by retraining the models, these figures can also be found in Appendix 5.3 in Figure 10. The loss and accuracy curves for all classifiers during training can be found in Appendix A.2.

#### 4.1.1 Inception-Net

As can be seen from the results presented in Figure 2 the created sweeps do not seem to indicate that only $\alpha$ influences the classifier output, in the $\beta$ sweeps we also see that changes in the classifier output are introduced. Furthermore it seems that the classifier is "tricking" itself and seems to not be able to classify correctly anymore. Before training the generative causal explainer, the classifier achieved a 99% accuracy as indicated in Appendix A.2. In the Figure 2 however, the classifier does not seem to be able to classify correctly.

The sweeps range from $[\alpha - 3, \alpha + 3]$ and $[\beta_i - 3, \beta_i + 3]$ with the middle column being examples drawn from MNIST. This holds for all figures that denote explanations of a certain classifier.

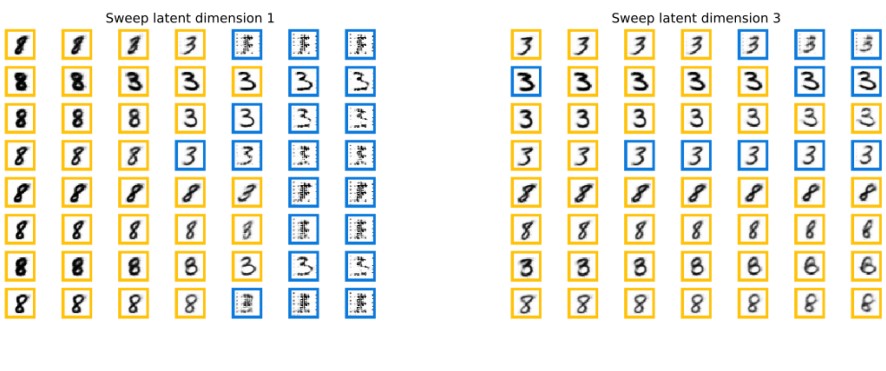

(a) $\alpha$ sweep           (b) $\beta_2$ sweep

Figure 2: Visualisations of learned latent factors for model using Inception-Net classifier. The colours indicate different outputs of the classifier. In this case yellow refers to a classification of 8 and blue refers to a classification of 3.

The generated examples, when visually examined, do not truly look like they are drawn from the underlying MNIST dataset. On the right hand side of the left plot of Figure 2 some of the generated images look more like random noise than actual digits.

#### 4.1.2 Res-Net

The results from the generative causal explainer using the Res-Net classifier seem more inline with the results obtained by the authors from the original paper. These results are presented in Figure 3. Visually we notice that the left plot,

which indicates the causal factor being changed, does indeed induce a change in classifications. Furthermore the right plot, which indicates noncausal factors, does not induce a change in classifier output. It does have one misclassification however in the 3rd row.

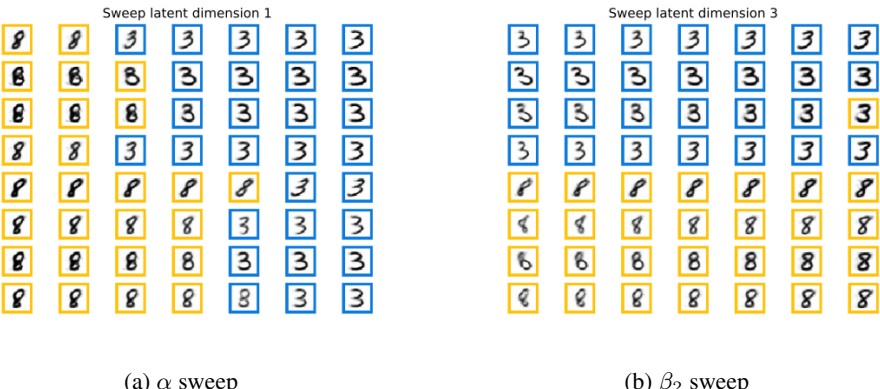

(a) $\alpha$ sweep
(b) $\beta_2$ sweep

Figure 3: Visualisations of learned latent factors for model using Res-Net classifier.

## 4.2 Hyperparameters

In this section we go over results that show the difficulties of the hyperparameter selection procedure described in section 3.3. The main three hyperparameters of the explanation model are K, L and $\lambda$, determining respectively the number of causal factors, the number of non-causal factors and the trade-off between causal influence and data fidelity terms of the objective function (2). All tests are performed on the fMNIST dataset, using the t-shirt, dress, and coat images. Classification is done with the pre-trained base network, as used by the authors. In the original paper, the authors give the optimal set of parameters with K=2, L=4 and $\lambda$=0.05. In our tests, training the explainer in each iteration of the parameter selection procedure is done with 3000 training steps.

As explained in section 3.3 the biggest difficulty of reproducing the parameter selection procedure is setting the correct criteria for selecting the optimal $\lambda$ value in step 2. Table 1 shows the results of selecting the optimal $\lambda$ when K=2 and L=4, the same latent parameters as the optimal set given by the authors. From this table it becomes clear that setting the criteria on a relative difference of 6% results in the optimal set K=2, L=4, $\lambda$=0.04, while setting the criteria on 5% result in the set K=2, L=5 and $\lambda$=0.07. Both these parameter sets are different than the optimal set given by the authors. Figure 4 and 5 show the visualised latent factors of both our obtained sets. For both sets we see the same phenomenon. The $\alpha$ sweep is correct, as it should change the output of the classifier. However, the $\beta$ sweeps in both figures seem to also change the output of the classifier, which is incorrect. This shows us two things, (1) the parameter selection procedure is difficult to operate. (2) the explanations by the latent factors are very sensitive to the selected hyperparameters.

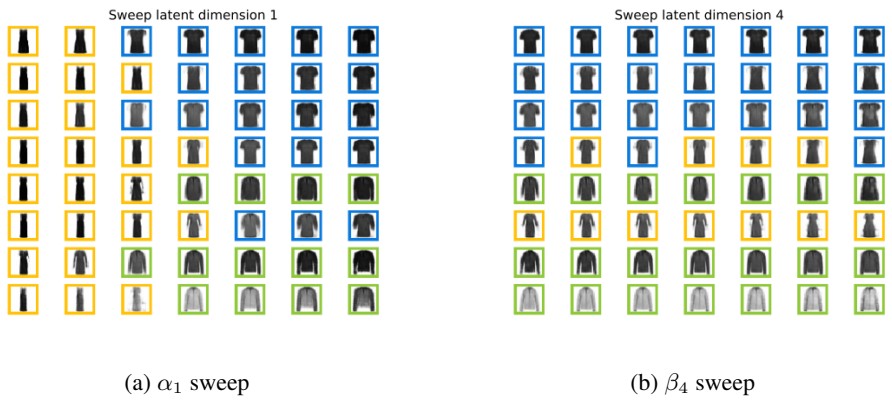

(a) $\alpha_1$ sweep
(b) $\beta_4$ sweep

Figure 5: Visualisations of learned latent factors for parameter set K=2, L4 and $\lambda$=0.07

| Configuration $\{K,L,\lambda\}$ | Relative distance to optimal D |
|---|---|
| $\{2, 4, 0.01\}$ | 17.60% |
| $\{2, 4, 0.02\}$ | 10.18% |
| $\{2, 4, 0.03\}$ | 6.71% |
| $\{2, 4, 0.04\}$ | 5.93% |
| $\{2, 4, 0.05\}$ | 5.54% |
| $\{2, 4, 0.06\}$ | 5.12% |
| $\{2, 4, 0.07\}$ | 4.88% |

Table 1: Results of optimising the $\lambda$ term for K=2 and L=4 in step 2 of the parameter selection procedure on the fMNIST dataset.

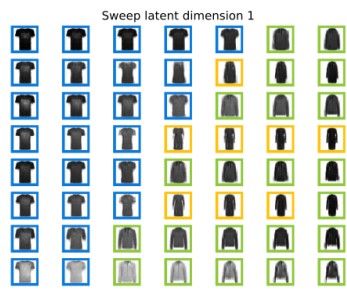

(a) $\alpha_1$ sweep

(b) $\beta_4$ sweep

Figure 4: Visualisations of learned latent factors for parameter set K=2, L4 and $\lambda$=0.04

### 4.3 CIFAR-10 Dataset

Three sets of classes were used in order to test the generalizability of the authors GCE. First we started with the two classes birds and planes.

The model provided by the author was not functioning correctly prior to adjustments. This is a result of the CIFAR-10 dataset introducing two additional color channels. Likewise we see that the plotting mechanism provided by the author was not able to process the three added color channels. This is why the visualisations (Fig. 15) were showing as binary images. Nevertheless, the classifier seems to work and the VAE is producing shapes but they are not really human interpretable yet.

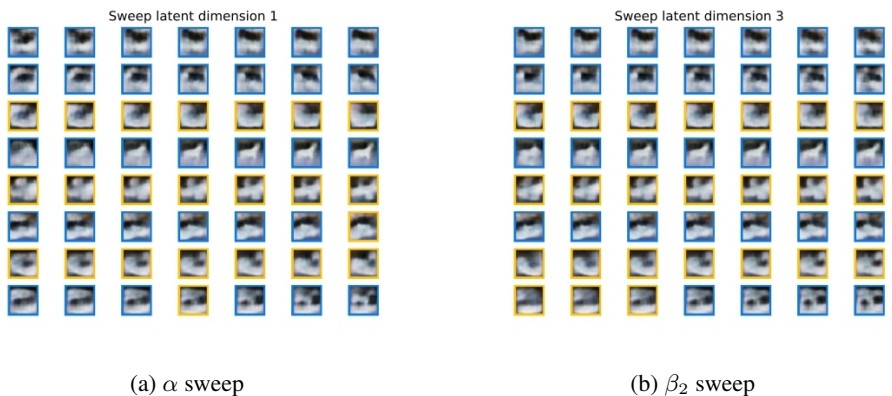

(a) $\alpha$ sweep

(b) $\beta_2$ sweep

Figure 6: Visualisations of learned latent factors for base model using CIFAR-10 (horses and trucks).

After redesigning the VAE and plotting mechanism to assure it accepts coloured (and grayscale) images the model was trained using the two classes horses and trucks. The results show (Fig. 6) that both the VAE and classifier took advantage of the extra information hidden in the colour channels.

Zooming into a single row of the $\alpha$ sweep (Fig. 7) shows that the VAE generates more human interpretable samples. Especially for the rows that contain horses. From left to right we see the change from a truck to a horse. Nevertheless we still notice that the classifier is not functioning perfectly. Adjusting the $\beta$ parameter should not influence the decisions made by the classifier. Still we see (Fig. 6b) that the classifications change when actually doing so.

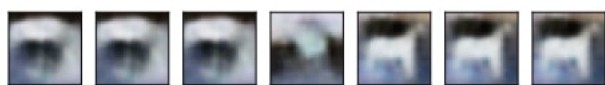

Figure 7: $\alpha$ sweep

Subsequently we wanted to challenge the classifier and VAE a bit further. Using the classes cats and dogs we propose images of animals with relatively similar structure.

The results show (Figure 16, in Appendix C) decreased performance for both the classifier and VAE. The reproduced samples do not show any human interpretable images. The classifier does function, but its accuracy is lower compared to using the horse and truck classes.

## 5 Discussion

The results of the reproducibility study show several things. Most importantly, with the original code written by the authors, it was possible to reproduce all results that the authors got during their research as shown in Appendix D.2.

However, while challenging their claims, a number of problems arised. Firstly, the claim that the method works with all black-box classifiers that are gradient-based and allow probabilities was not possible to validate with this research. Testing with more complex convolutional classifiers showed a deterioration of the explanations generated by the GCE. The more complex the classifier, the more the explanations of the GCE worsened. The most probable explanation for this decline in result quality is that with the growing complexity of the models, the decision boundary for the classes also grows more complex. The base generative model is not complex enough to approach this decision boundary and generate valid explanations.

Secondly, during testing of the hyperparameter selection method, it was found that the GCE is heavily influenced by other factors, e.g. classifier choice or different datasets, showing that the method is not robust and needs extensive parameter tuning to be usable.

Finally, the generalizability of the explanation method was tested and shown to be lacking. The complexity of the CIFAR-10 dataset resulted in poor loss values and an inadequate representation of the latent space. The explanations that were generated were supposedly correct but not human-interpretable, as the classifier often changed its output even though no changes in sweep could be detected by the human eye.

Putting this all together we can conclude that the authors work is reproducible to some extent. The figures provided in the original paper are able to be reproduced. The difficulty in applying the method however may indicate that the work is not easily generalizable or usable in real-world scenarios. Time constraints during the research for this paper may also have had an effect on our results, as we were not able to extensively test more complex VAE architectures to use as a generative causal explainer.

### 5.1 What was difficult

The initial code provided by the author was of chaotic structure and did not function as intended. When following the included instructions the code did not reproduce the results as listed in the original paper. Reorganising the code and fixing the underlying bugs took more time than anticipated. Subsequently to the author responding to our changes these complications were less problematic.

### 5.2 What was easy

After a rigorous update of the base code provided to us by the authors, reproducing the original authors' results was trouble-free. We only needed to specify what result we wanted to show and the code worked as intended.

### 5.3 Communication with original authors

The authors established contact with our group after seeing our fork of their code. The authors cleaned up their code for us to work with and kept in contact with us during the entire project. For instance, when problems arised during testing with other classifiers, one of the authors helped us figure out one of the possible reason for these problems. Moreover, every question we had could be posed to the authors without issues.

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

# Appendix

## A    Classifiers

### A.1    Classifier Descriptions

#### A.1.1    Base network

This network, created by the authors of the original paper, used two convolutional layers which scaled the input image up to 64 channels, both of these layers were followed by a rectified linear unit (ReLU) activation layer. After the scaling up of the image a maximum pooling layer using a kernel of 2x2 was used. Following the pooling layer a dropout layer with a dropout percentage of 0.25 was applied, followed by 2 fully connected layers which downscales the input down to the amount of classes the data contains. In between the two fully connected linear layers another dropout layer with a dropout percentage of 0.5 was used. After the first linear layer a ReLU activation layer was used, after the second linear layer a softmax activation layer was used to transform the data to a probability distribution for inference.

#### A.1.2    ResNet

ResNet is a deep neural network based on the idea of residual connections. Residual connections allow for stable gradient propagation through a network. Residual connections model $x_{l+1} = x_l + F(x_l)$ instead of the more traditional $x_{l+1} = Fx_l$. The addition of $x_l$-term guarantees stabler gradient propagations [7].

The ResNet architecture contains multiple residual blocks (visually indicated in Figure 8a) stacked on top of each other to create deep networks. This block is visually shown in Figure 8a.

We used a smaller version of the original ResNet proposed in [7]. For our model we stacked 3 of these residual blocks. Our residual blocks used convolutional layers with a kernel of 3x3 and a padding of 1. Before feeding the input images into these blocks first the images were scaled up to have 16 colour channels using a convolutional layer and a batch normalization layer followed by an activation function layer. The output of these blocks are then fed into an output network which uses an average pooling layer and a linear layer to convert the output to the correct number of possible classifications. We used the ReLU activation function as our non-linearity, and used a softmax function on our output to convert it to a probability distribution.

#### A.1.3    DenseNet

DenseNet is an architecture that enables very deep neural networks by using residual connection. But instead of modeling the difference between layers using the residual connections the model considers residual connections as a way to reuse features across layers. This allows the model to remove redundant features. A general DenseNet architecture is shown in Figure 8c.

In our case we implemented the DenseNet by defining three modules, a DenseLayer, a DenseBlock, and a Transition-Layer. A DenseLayer is one of the smaller squares in the figure, while a DenseBlock is multiple DenseLayer stacked on top of each other. A TransitionLayer is the last layer and transforms the dimensionality of the features as they flow through the model. In our implementation one DenseLayer contains a 1x1 convolution followed by a subsequential 3x3 convolution. The output channels of these convolutions are concatenate to the original input. Furthermore we apply a batch normalizations throughout the layer to stabilize training. The non-linearity we use is the ReLU activation function. A DenseBlock in our implementation consists of 3 DenseLayers followed by a TransitionLayer. In our model we stacked one of these DenseBlocks due to the relative simplicity of the task. After the data was pulled through the block the output was transformed using an average pooling layer and a linear layer [8].

#### A.1.4    Inception

The GoogleNet, code named Inception, stacked multiple so-called convolutional blocks on top of each other. These blocks are called Inception blocks. An Inception block applies four convolution layers on the same input: a 1x1, 3x3, 5x5, and a max pool layer. The outputs of these layers are then concatenated and passed on to the next block. In Figure 8b an overview of a general Inception block is shown [9].

Since the original network was proposed to work with images of size 224x224, and the MNIST dataset only contains images of size 28x28, we use a smaller down-scaled version of the network. First the input image was scaled up to 64 channels using a convolutional layer followed by a batch normalization layer for stability. After the transformation, our network used only one Inception block, followed by a maximum pooling layer, an average pooling layer, and

a linear layer to transform the output to the number of classes needed for classification. The non-linear activation function we used after each convolutional layer was the ReLU activation layer. Only the output layer used a softmax layer to transform the output into a probability distribution.

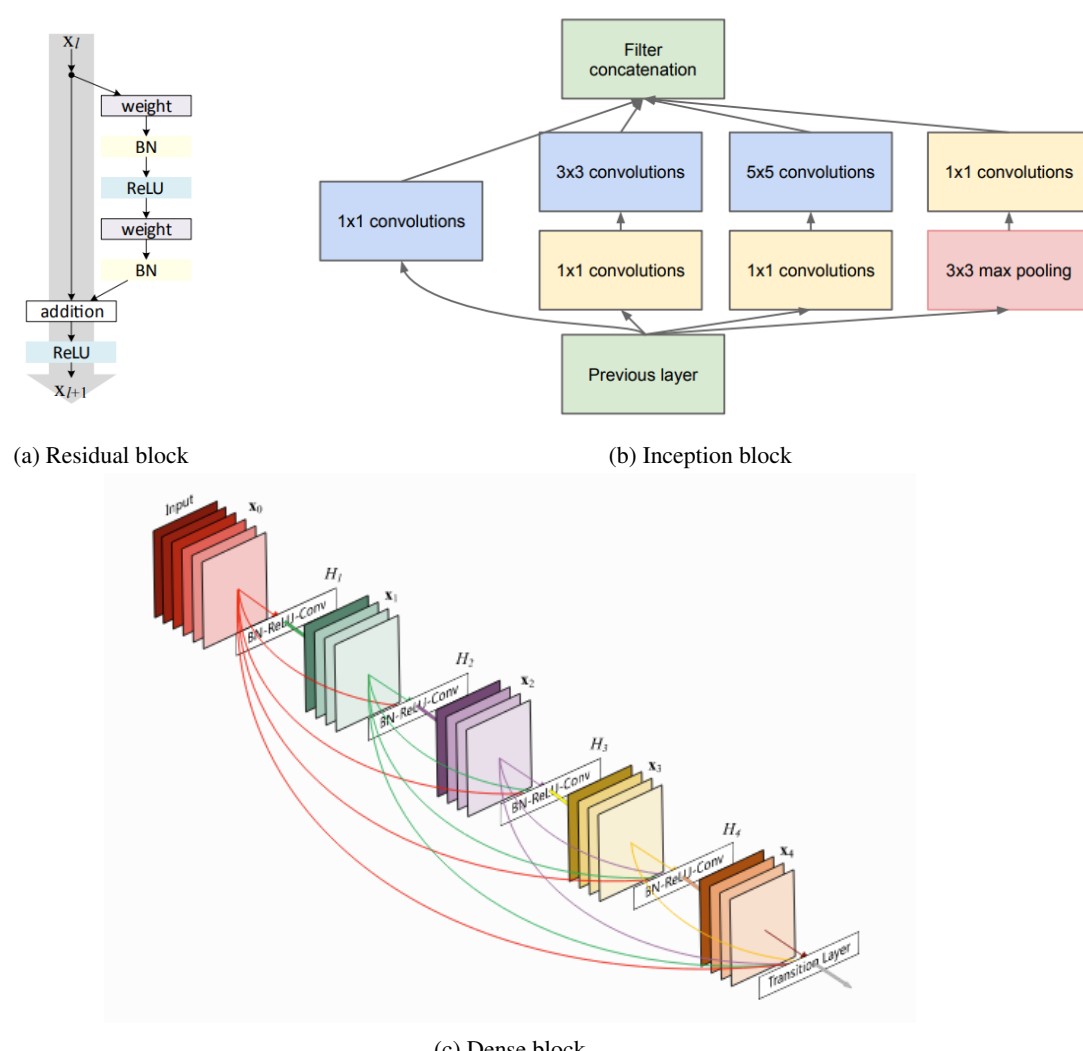

(a) Residual block          (b) Inception block

(c) Dense block

Figure 8: Classifier blocks [7, 9, 8]

## A.2   Loss/accuracy curves during training

Curves gathered during training of the classifiers, the x-axis denotes the epoch and the y-axis the loss or accuracy during training. All models reach 100% accuracy fairly quickly, this has to do with the fact that the data fed to the models are only consisting of 2 classes. The 2 classes being 3 and 8 from the MNIST dataset. Our 3 classifiers, DenseNet, ResNet, and InceptionNet all achieve a very high performance of 100% while the base network achieved a relatively lower accuracy of 96%.

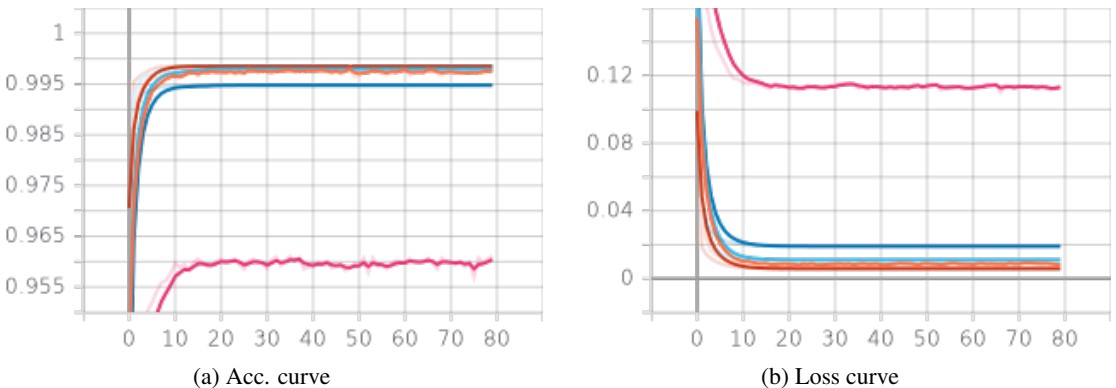

(a) Acc. curve                    (b) Loss curve

Figure 9: Loss/Accuracy curves during training of classifiers

 ## A.3   Explanations

 ### A.3.1   Base

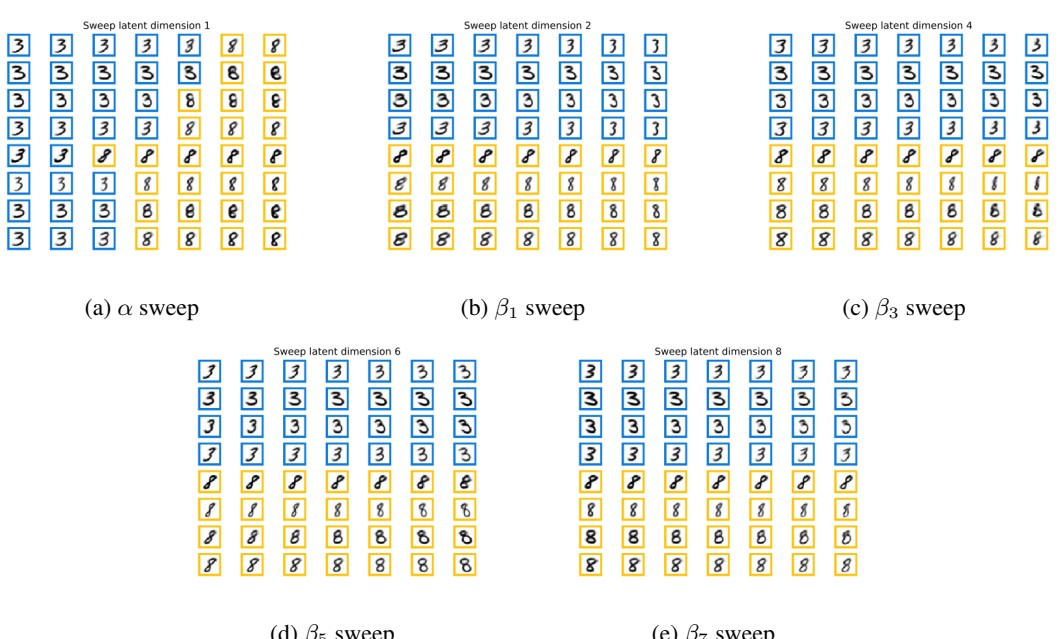

(a) $\alpha$ sweep              (b) $\beta_1$ sweep              (c) $\beta_3$ sweep

(d) $\beta_5$ sweep              (e) $\beta_7$ sweep

Figure 10: Visualisations of learned latent factors for model using the base classifier provided by the authors.

 ### A.3.2 DenseNet

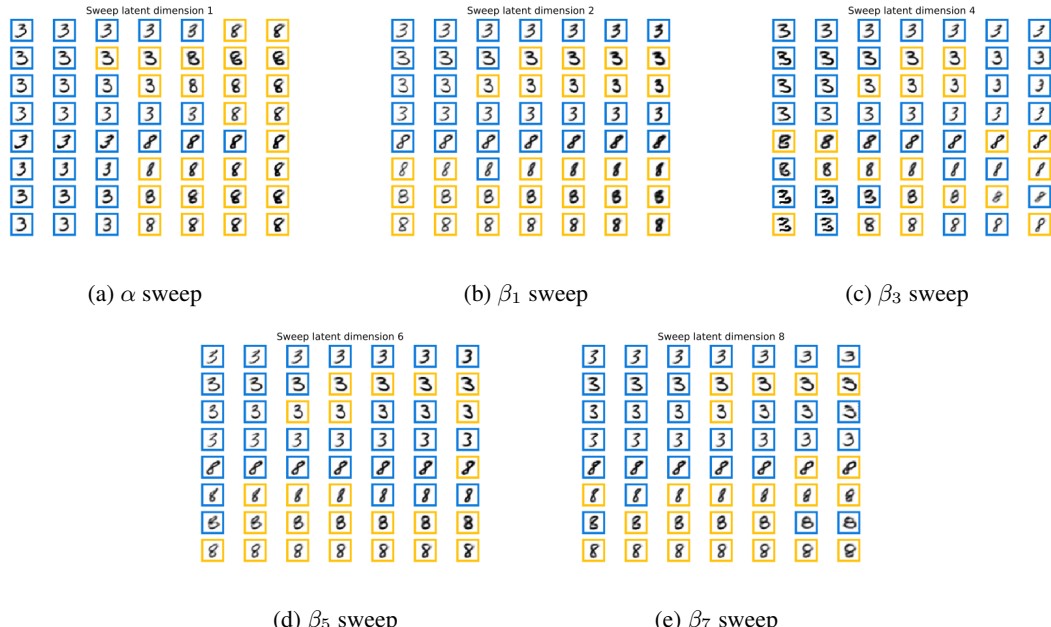

(a) $\alpha$ sweep  (b) $\beta_1$ sweep  (c) $\beta_3$ sweep

(d) $\beta_5$ sweep  (e) $\beta_7$ sweep

Figure 11: Visualisations of learned latent factors for model using DenseNet classifier.

 ### A.3.3 ResNet

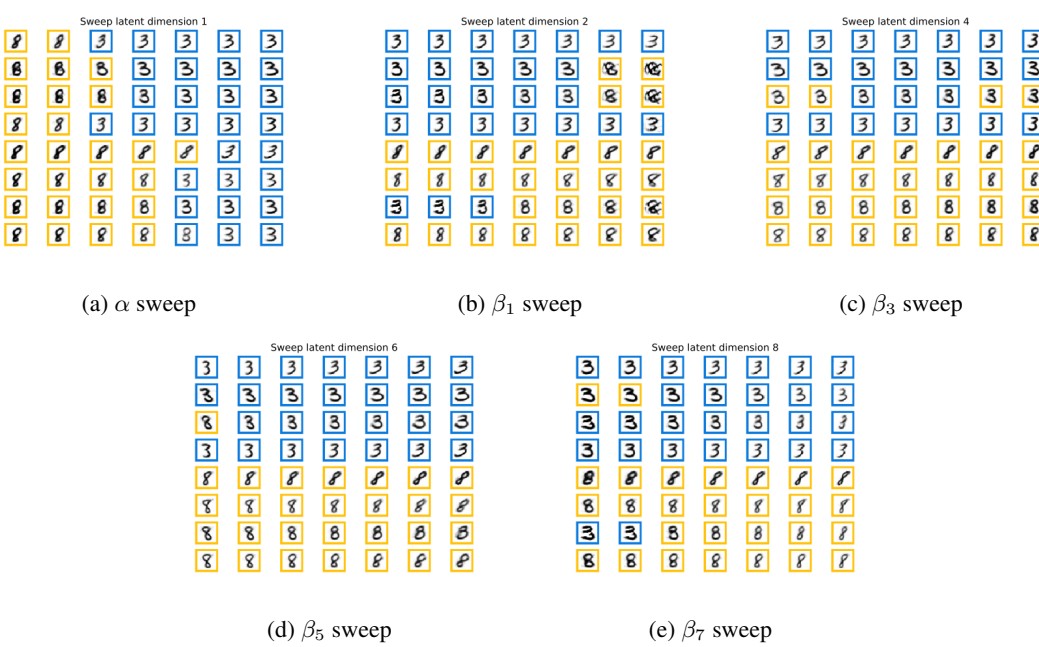

(a) $\alpha$ sweep  (b) $\beta_1$ sweep  (c) $\beta_3$ sweep

(d) $\beta_5$ sweep  (e) $\beta_7$ sweep

Figure 12: Visualisations of learned latent factors for model using ResNet classifier.

 **A.3.4 InceptionNet**

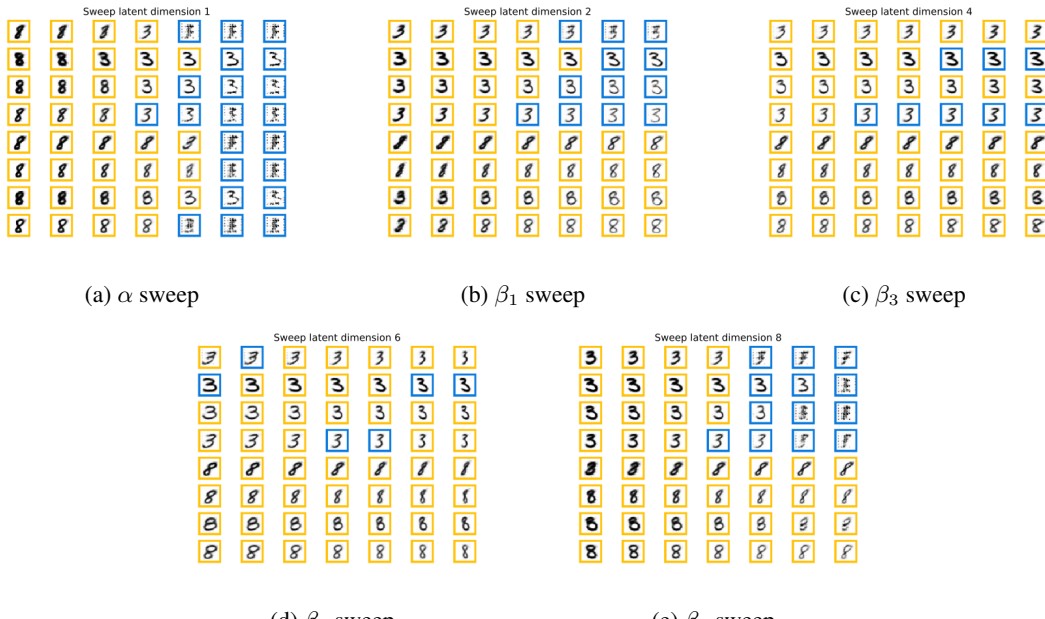

(a) $\alpha$ sweep        (b) $\beta_1$ sweep        (c) $\beta_3$ sweep

(d) $\beta_5$ sweep        (e) $\beta_7$ sweep

Figure 13: Visualisations of learned latent factors for model using InceptionNet classifier.

**B   Hyperparameter selection**

**B.1   High volatility of data similarity score**

In this section we show the complementary results of implementing the hyperparameter selection procedure. As mentioned in section 3.3, step 1 of the procedure optimises data similarity between the learned data distribution and the original data distribution measure by $\mathcal{D}(p(\alpha, \beta), p(X))$ in the objective function. As described by the authors, step 1 is finished when the data similarity plateaus. However, we found that the data similarity that is measured was too volatile to easily compare across runs. This is shown in figure 14 in which the value for $\mathcal{D}$ is plotted per training step.

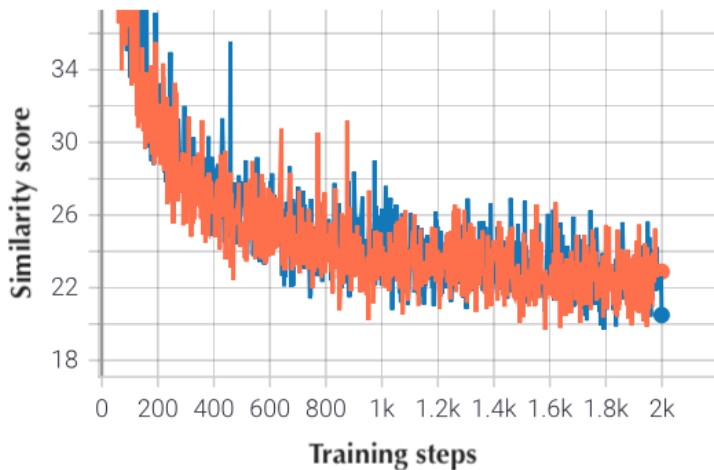

Figure 14: Complementary results of parameter selection step 1. Showing the high variance of $\mathcal{D}$ results while training. The x-axis shows the training steps, the y-axis shows the similarity score $\mathcal{D}$. These are measurements of two runs on the fmnist dataset, (orange) K=0, L=5, $\lambda$=0.01 and (blue) K=0, L=6, $\lambda$=0.01.

# C  Datasets

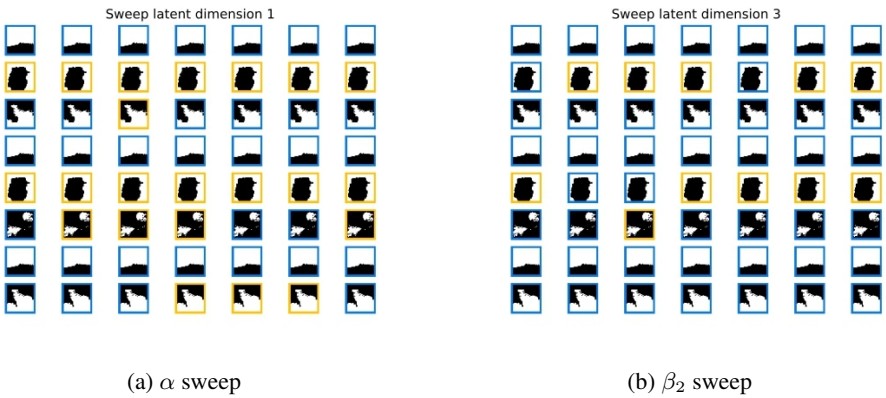

(a) $\alpha$ sweep  (b) $\beta_2$ sweep

Figure 15: Visualisations of learned latent factors for base model using CIFAR-10 (birds and planes).

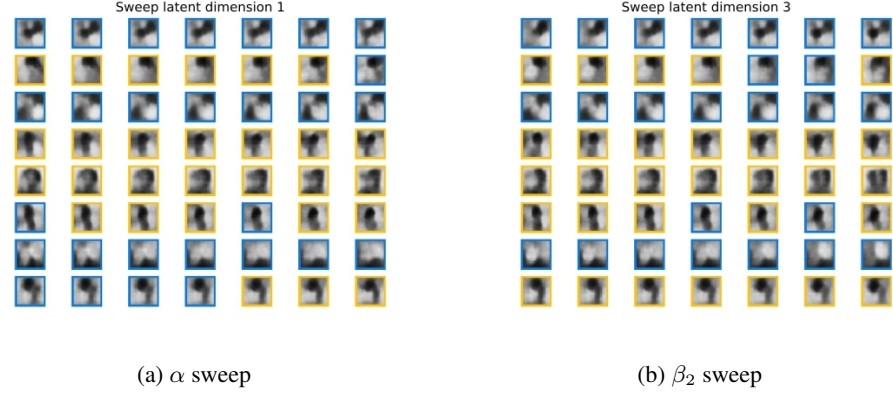

(a) $\alpha$ sweep  (b) $\beta_2$ sweep

Figure 16: Visualisations of learned latent factors for base model using CIFAR-10 (horses and trucks).

# D  Reproductions

We reproduced the figures presented in the original paper by using the author's provided code. We used the instructions provided on their repository to generate these images. We re-trained all the networks that we could when generating these images, and found the figures generated by the scripts to be identical to the ones provided in the paper.

 **D.1 Figure 3**

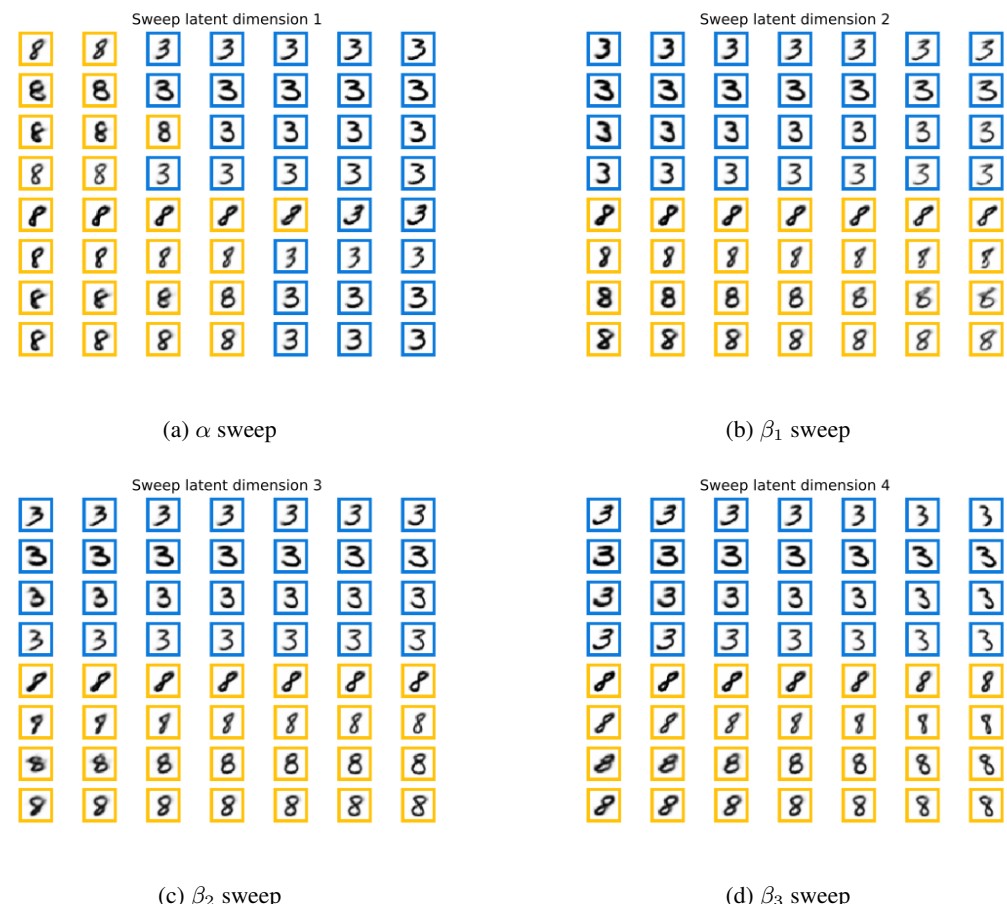

(a) $\alpha$ sweep

(b) $\beta_1$ sweep

(c) $\beta_2$ sweep

(d) $\beta_3$ sweep

Figure 17: Reproduction of visualisation of learned latent factors provided in Figure 3 of the original paper.

 **D.2 Figure 4**

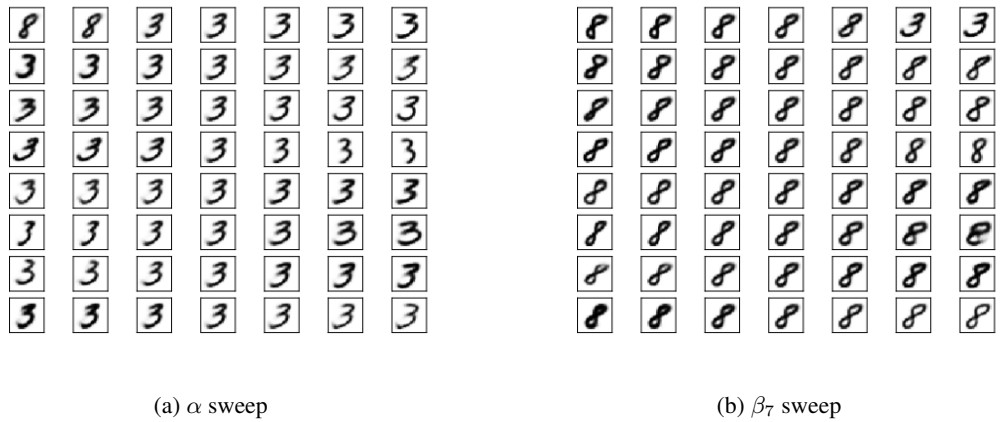

(a) $\alpha$ sweep

(b) $\beta_7$ sweep

Figure 18: Reproduction of visualisation of learned latent factors provided in Figure 4 of the original paper. Showing how the explanations are only causally influenced by $\alpha$

 **D.3   Figure 5**

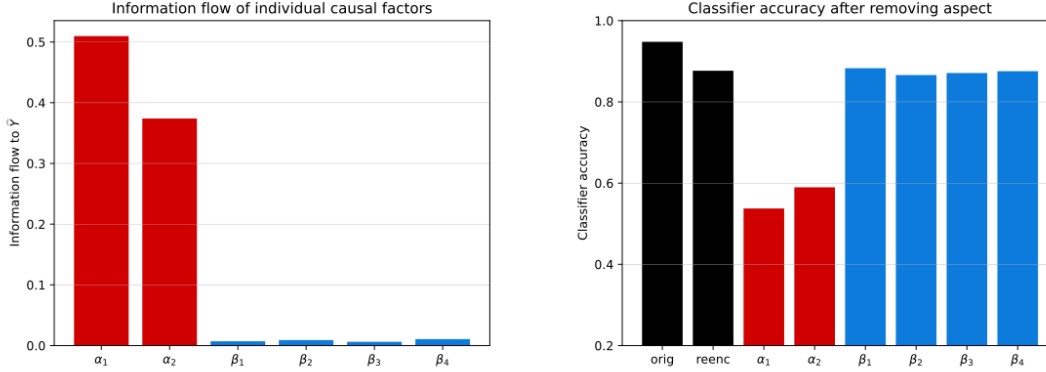

Figure 19: Reproduction of visualisation of the Figures 5(a) and 5(b)

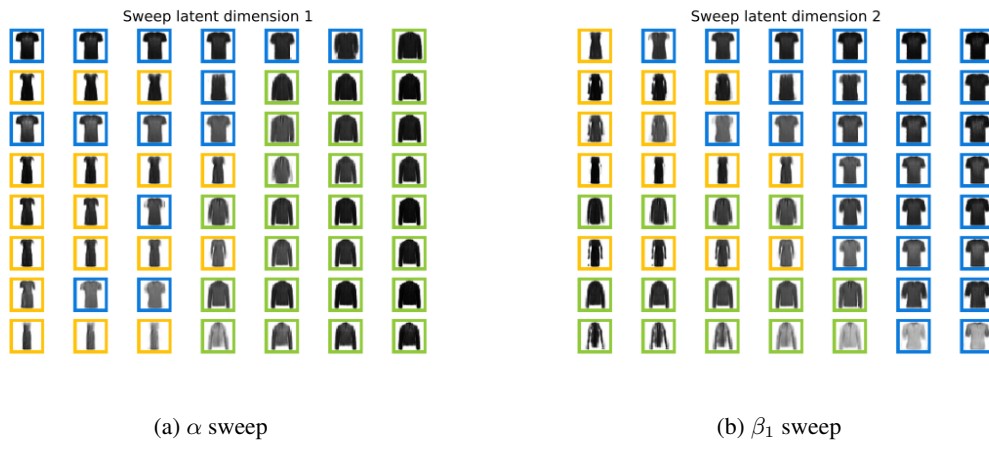

(a) $\alpha$ sweep

(b) $\beta_1$ sweep

Figure 20: Reproduction of visualisation of the Figures 5(c) and 5(d)

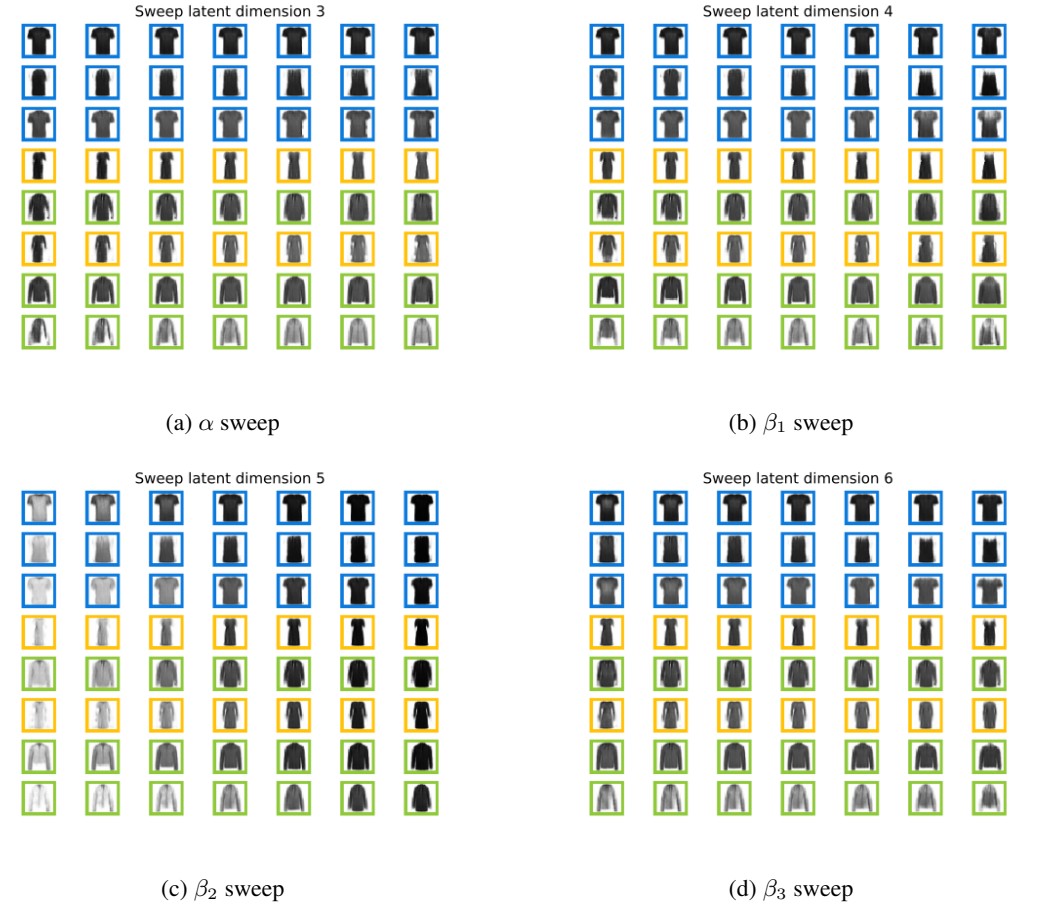

(a) $\alpha$ sweep

(b) $\beta_1$ sweep

(c) $\beta_2$ sweep

(d) $\beta_3$ sweep

Figure 21: Reproduction of visualisation of learned latent factors provided in Figure 5 of the original paper. Here we show more $\beta$ parameters and how they don't affect the output of the classifier.

