# OpenReview forum: "[Re] Generative causal explainers of black-box classifiers"
_ML_Reproducibility_Challenge/2020 — Reject_

### Official Review · AnonReviewer3 · 2021-03-02
**RE: [Re] Generative causal explainers of black-box classifiers**

**Rating:** 6
**Confidence:** 4

**Review:**

In this paper, the authors reported results from the reproducibility study of [1].

Pros:
1), successfully reproduced the results reported in the original paper
2), went beyond the empirical study in the original paper by testing the method with networks of more complicated architectures and additional datasets of increased learning difficulty to test the scope of the usability of the method.
3), performed study of searching for hyperparameters: K, L, \lambda
4), good communications with authors of original paper

Cons:
1), the writing needs substantial improvement
2), study of using more complicated generative model is needed for more complicated classifiers or learning tasks. The proposed method in the original paper may still perform well using more complicated generative models for cases where there is increased complicity in the learning task and classifier.


**Familiar With The Original Paper:**

I have read the original paper

**Reproducibility Summary:**

Report has summary

---

### Official Review · AnonReviewer2 · 2021-03-02
**Some good seeds but not quite mature yet**

**Rating:** 5
**Confidence:** 4

**Review:**

The report details reproduction of the paper "Generative causal explanations of black-box classifiers" by O’Shaughnessy et al. The authors tried to do all the right things and even go beyond only reproducibility but also to extend the original experiments but some parts are rushed and some conclusions drawn are questionable.

Reproducibility Summary: is provided but it is quite short and cover the main points of the reports only at the very high level. It could have been extended to include more specific details.

Scope of reproducibility: is clearly stated and is followed in the later text.

Code: the authors used the code provided by the original authors of the paper. They mentioned several issues with the code which they had to overcome (sometimes with the help of the original authors). The release code by the authors of the report looks quite readable and well organised.

Communication with original authors: The report mentions communication with the original authors but mostly in terms of working with their code. However, the authors emphasise that the main issue with reproducibility was selection of hyperparameters and no discussion of that with the original authors is mentioned.

Hyperparameter Search: the authors emphasise that hyperparameter search method provided in the original paper is not very clear, nor it is well motivated, nor different values are checked in the original paper. Moreover, the authors claim that the method from their experiments is quite sensitive to this choice of hyperparameters. However, although claimed to conducted experiments on hyperparameter search, the authors only report results for tuning only 1 of the 3 main hyperparameters. Even with these experiments the provided results do not seem to confirm the authors' claim on high sensitivity. I may be missing something but the results of figures 4 and 5 look consistent enough for me.

Ablation Study: no ablation study has been conducted

Discussion on results: the report does have this discussion and clearly states which parts of their reproducibility study was easy and difficult. However, some drawn conclusions by the authors are questionable. For example, the authors claim that from their experiments the model is highly sensitive to selection of hyperparameters. Considering that this is true (questionable on its own (see above)), then the authors claim that the model is not generalisable to a more difficult dataset, however, they do not report at all on hyperparameter selection for this dataset. From this it might be the case that this is still the same issue of sensitivity to hyperparameters rather than the additional issue with generalisability.

Recommendations for reproducibility: are provided to some extent. The main recommendation is considering the hyperparameter selection procedure.

Results beyond the paper: the authors go in several directions beyond the paper. They explore different base classifiers that the proposed method from the original paper should explain and they test the model on a more difficult dataset.

Overall organisation and clarity: the report is mostly well written and easy to follow. There are a number of minor typos.

Some specific points:
1.	Line 29, “parameter selection of the paper” – bad wording
2.	Line 47, DAG is not introduced
3.	Lines 118-119, “we introduce the following two variants” which follows only by one option
4.	Lines 135-136, “These values …” should better be placed for the sentence “As described in Section 2.1.3” as the method from Figure 1 only determines values for K, L, and lambda and not the number of samples N_alpha and N_beta
5.	Figure 2 requires more explanation about set up of the experiment and what different colours mean
6.	Section 4.3 – everything should be adjusted to 3 input channels at once and there is no need to compare results without doing so

Minor:
1.	Line 10, “of the of the” – the second of the is redundant
2.	Line 83, “traditioanally” -> “traditionally”
3.	Both “grayscale” and “greyscale” is used in the text. It should be consistent
4.	Line 117, “i.e.” is redundant
5.	Line 123, “. Which” -> “, which”
6.	Line 186, “In this table becomes” -> “From this table it becomes”
7.	Line 196 – missing full stop at the end of the sentence
8.	Line 202, “Posterior to redesigning” -> “After redesigning”
9.	Line 206, “Especially for the horses” – unfinished sentence
10.	Line 217, “A number” -> “a number”
11.	Line 240, “pragmatic” – it seems the authors meant something else


**Familiar With The Original Paper:**

I have not read the original paper

**Reproducibility Summary:**

Report has summary

---

### Official Review · AnonReviewer1 · 2021-03-04
**Review for [Re] Generative causal explainers of black-box classifiers**

**Rating:** 7
**Confidence:** 3

**Review:**

This paper aims to reproduce and examine the claims made in the "Generative causal explanations of black box classifiers" paper. The authors approach this problem by examining three aspects of behavior of the proposed model:
1. Reproducibility of the results presented in the original paper.
2. Sensitivity to alternative black box models.
3. Sensitivity to the choice of hyperparameters.
The authors find positive evidence of (1). However, they find a decay in performance with more complex models and a lack of robustness with respect to the choice of hyperparameters.

Overall, I felt this was a solid reproduction. On the positive side: the authors do well to push the proposed methodology outside of the aspects presented within the paper, and I found the findings to be informative.

On the less positive side: (a) Ideally, I would have liked to seen at least a set of hypotheses summarizing the explanation of the degradation of performance. (b) Some of the writing could use a little more fleshing out, the descriptions in the beginning of the text are spars and make it difficult to have full context without having read the prior paper within a very short window.

**Familiar With The Original Paper:**

I have read the original paper

**Reproducibility Summary:**

Report has summary

---

### Decision · Program_Chairs · 2021-03-31

**Decision:**

Reject

**Comment:**

Overall reviews and/or the paper content not good enough for the AC to recommend to the journal.